# Synthesize, Execute and Debug: Learning to Repair for Neural Program Synthesis

**Kavi Gupta**
UC Berkeley
kavi@berkeley.edu

**Peter Ebert Christensen** *
Technical University of Denmark
pebch@dtu.dk

**Xinyun Chen***
UC Berkeley
xinyun.chen@berkeley.edu

**Dawn Song**
UC Berkeley
dawnsong@cs.berkeley.edu

## Abstract

The use of deep learning techniques has achieved significant progress for program synthesis from input-output examples. However, when the program semantics become more complex, it still remains a challenge to synthesize programs that are consistent with the specification. In this work, we propose SED, a neural program generation framework that incorporates synthesis, execution, and debugging stages. Instead of purely relying on the neural program synthesizer to generate the final program, SED first produces initial programs using the neural program synthesizer component, then utilizes a neural program debugger to iteratively repair the generated programs. The integration of the debugger component enables SED to modify the programs based on the execution results and specification, which resembles the coding process of human programmers. On Karel, a challenging input-output program synthesis benchmark, SED reduces the error rate of the neural program synthesizer itself by a considerable margin, and outperforms the standard beam search for decoding.

## 1 Introduction

Program synthesis is a fundamental problem that has attracted a lot of attention from both the artificial intelligence and programming languages community, with the goal of generating a program that satisfies the given specification [17, 10]. One of the most popular forms of specifications is to provide a few examples of program inputs and the desired outputs, which has been studied in various domains, including string manipulation [10, 5] and graphics applications [8, 7].

There has been an emerging line of work studying deep learning techniques for program synthesis from input-output examples [5, 29, 2, 3]. Some recent work demonstrate that a large performance gain can be achieved by properly leveraging the execution results to guide the program synthesis process [23, 27, 31, 3, 7]. In particular, in [3, 31, 7], they propose to make predictions based on the intermediate results obtained by executing the partially generated programs. This scheme is well-suited for sequential programming tasks [31, 7]; however, for programs that are loop-containing, the execution results of partial programs are not always informative. For example, in [3], to execute a partial while loop, its body is only executed once [3], making it effectively equivalent to an if statement. Furthermore, existing work on program synthesis generate the entire program from scratch without further editing, even if the predicted program is already inconsistent with the input-output specification and thus incorrect. On the contrary, after observing wrong program outputs,

human programmers would go through the written code and attempt to fix the program fragments that cause the issue.

Inspired by the trial-and-error human coding procedure, we propose SED, which augments existing neural program synthesizers with a neural debugger component to repair the generated programs. Given the input-output specification, SED first *synthesizes* candidate programs with a neural program generation model. Next, SED *executes* the predicted programs and see if any of them satisfies the input-output examples. If none of them does, SED proceeds into the *debugging* stage, where it selects the most promising program from the candidates, then generates editing operations with a neural network component acting as the debugger. The neural program debugger iteratively modifies the program until it passes the specification or reaches the maximal number of editing iterations. Besides the syntactic information of candidate programs as token sequences, our debugger also leverages the semantic information provided by their execution traces, which facilitates it to fix the semantic errors.

We evaluate SED on Karel benchmark [2, 4], where the programs satisfying input-output examples could include control flow constructs such as conditionals and loops. With different choices of the neural program synthesis model, SED consistently improves the performance of the synthesizer itself by a considerable margin. Meanwhile, when the synthesizer performs the greedy decoding and only provides a single program for editing, SED outperforms the standard beam search applied to the synthesizer, which further demonstrates the effectiveness of our SED framework.

## 2   Problem Setup

In this section, we present the setup of the input-output program synthesis problem, which is the main focus of this work. We will also introduce the program repair problem handled by our neural debugger component.

**Program synthesis from input-output examples.** In a standard input-output program synthesis problem [5, 2], the synthesizer is provided with a set of input-output pairs $\{(i_k, o_k)\}_{k=1}^{K}$ (or $\{IO^K\}$ in short), which serves as the *specification* of the desired program semantics. Let the ground truth program be $P^\star$, the goal of the program synthesizer is to generate a program $P$ in a domain-specific language (DSL) $\mathcal{L}$, so that for any valid input $i'$, $P(i') = P^\star(i') = o'$. In practice, besides $\{IO^K\}$, usually another set of "held out" input-output examples $\{IO^{K_{test}}\}_{test}$ is generated to verify the equivalence between the generated and ground truth programs, which could be imprecise due to the typically small test set.

**Program repair with input-output specification.** In our program repair setting, besides the same input-output specification given for the synthesis problem above, a buggy program $P'$ is also provided, which is inconsistent with the specification. The goal is to perform editing operations, so that the modified program becomes correct.

Intuitively, program repair is no more difficult than the program synthesis problem, as the editing process can completely remove the provided program $P'$ and synthesize a new program. However, it is usually beneficial to utilize $P'$, especially when it is close to a correct program [25].

**Karel domain.** The Karel programming language goes back to the 1980s as an introductory language for Stanford CS students [21], and some recent work have proposed deep learning approaches with this domain as a test bed [2, 23, 3, 25, 24]. A program in this language describes the movements of a robot inside a 2D grid world, and we present a sample program in Figure 1. In the grids, the arrows represent the robot, the grey cells are obstacles that can not be manipulated, and the dots are markers. Besides an action set for the robot, Karel language also includes control flow constructs, i.e., `if`, `repeat` and `while`. The full grammar specification is discussed in Appendix B.

## 3   SED: Synthesize, Execute and Debug

In this section, we demonstrate SED, which learns to debug for neural program synthesis. In the synthesis phase, SED uses a neural program synthesizer to produce candidate programs. When the programs do not satisfy the specification according to the execution results, a debugging phase is required before providing the final program for evaluation. Figure 1 provides an example of how

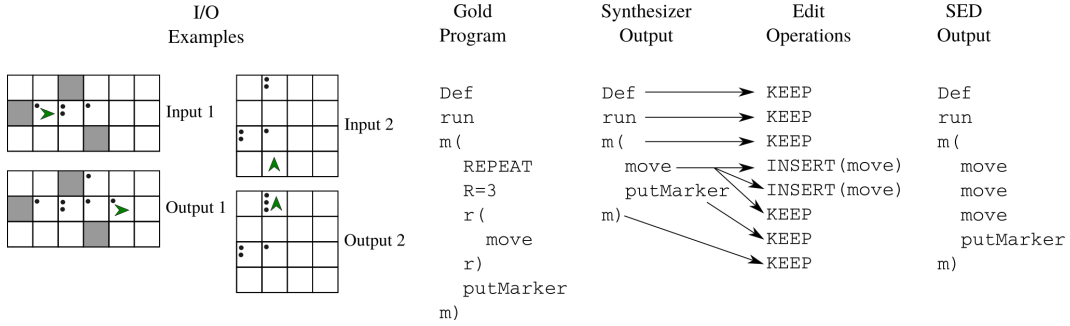

Figure 1: A sample debugging process of SED. Given the input-output examples, the synthesizer provides a wrong program that misses the repeat-loop in the ground truth. Our debugger then performs a series of edits, which results in a correct program. Note that the `INSERT` operation does not advance the pointer in the input program, so several edits are applied to the `move` token.

SED works. In the following, we first present the neural network architecture, then describe the training and inference procedures.

## 3.1 Synthesizer

Our SED framework is largely agnostic to the choice of the synthesizer, as long as it achieves non-trivial prediction performance, thus it is beneficial to leverage its predicted programs for the debugger component. In particular, SED is compatible with existing neural program synthesis models that largely employ the encoder-decoder architectures [2, 3, 5]. A common model architecture for input-output program synthesis includes an encoder to embed the input-output pairs, which could be an LSTM for string manipulation tasks [5], or a convolutional neural network for our Karel task [2, 3, 23]. Then, an LSTM decoder generates the program based on the input embedding.

## 3.2 Debugger

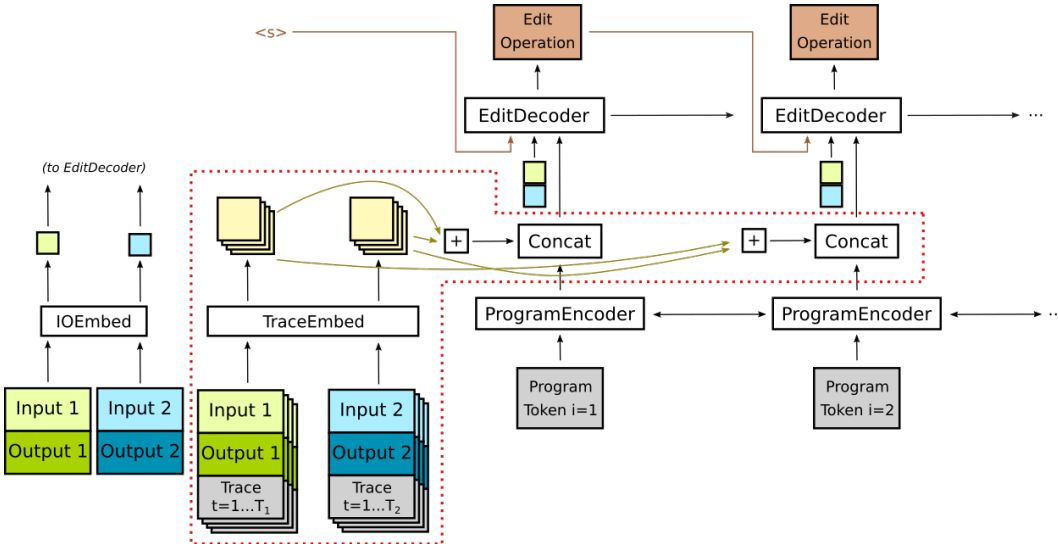

Figure 2: The neural debugger model in SED. The encoder consists of three parts: (1) `IOEmbed` for I/O embedding; (2) `TraceEmbed` that convolves the traces with their corresponding I/O pairs; and (3) `ProgramEncoder` that jointly embeds each program token with its corresponding execution steps in the trace. `EditDecoder` is used for generating edits. We outline our proposed `TraceEmbed` component in red dots, which is the key architectural difference compared to [25]. *Note: We use the light blue and green squares to indicate that the same values are passed into the edit decoder at every step.*

We present the debugger architecture in Figure 2. We follow previous work for Karel domain to use a convolutional neural network for I/O embedding, a bi-directional LSTM to encode the program for debugging, and an LSTM to sequentially generate the edit operation for each input program token [25]. The debugger supports 4 types of edit operations: KEEP copies the current program token to the output; DELETE removes the current program token; INSERT[$t$] adds a program token $t$; and REPLACE[$t$] replaces the current program token with $t$. Therefore, the total number of edit operations is $2|V| + 2$, where $|V|$ is the Karel vocabulary size. For KEEP, REPLACE and DELETE, the LSTM moves on to process the next program token after the current edit operation, while for INSERT, the next edit operation still based on the current program token, as shown in Figure 1.

The input program serves as a cue to the desired syntactic structure; however, it may not be sufficient to reveal the semantic errors. Motivated by the breakpoint support in Integrated Development Environments (IDEs) for debugging, we propose an execution trace embedding technique and incorporate it into the original debugger architecture, as highlighted in Figure 2. Specifically, we first execute the input program on each input grid $i_u$, and obtain the execution trace $e_{u,0}, ..., e_{u,t}, ..., e_{u,T}$, where $u \in \{1, 2, ..., K\}$. For each state $e_{u,t}$, we use a convolutional neural network for embedding:

$$te_{(u,t)} = \texttt{TraceEmbed}([e_{u,t}; i_u; o_u]) \tag{1}$$

Where $[a; b]$ means the concatenation of vectors $a$ and $b$.

To better represent the correspondence between each program token and the execution states it affects, we construct a bipartite graph $E \subseteq G \times I$, where $G$ is the set $\{(u, t)\}$, and $I$ is the set of program token indices. We set $((u, t), i) \in E$ iff the program token $p_i$ was either executed to produce $e_{u,t}$; or $p_i$ initiates a loop or conditional, e.g., repeat, and the body of that loop or conditional produces $e_{u,t}$ when executed. For each program token $p_i$, we compute a vector representation of its related execution states, which is the mean of the corresponding embedding vectors:

$$q_i = \frac{1}{|\{(u, t) : ((u, t), i) \in E\}|} \sum_{(u,t):((u,t),i)\in E} te_{(u,t)} \tag{2}$$

Finally, the program token representation fed into the edit decoder is $ep'_i = [ep_i; q_i]$, where $ep_i$ is the original program token embedding computed by the bi-directional program encoder.

### 3.3 Training

We design a two-stage training process for the debugger, as discussed below.

**Stage 1: Pre-training with synthetic program mutation.** We observe that if we directly train the debugger with the predicted programs of the synthesizer, the training hardly makes progress. One main reason is because a well-trained synthesizer only makes wrong predictions for around $15\% - 35\%$ training samples, which results in a small training set for the debugger model. Although the synthesizer could produce a program close to one that satisfies the input-output specification, it is mostly distinct from the annotated ground truth program, as indicated in our evaluation. Therefore, we build a synthetic program repair dataset in the same way as [25] to pre-train the debugger. Specifically, for each sample in the original Karel dataset, we randomly apply several mutations to generate an alternative program $P'$ from the ground truth program $P$. Note that the mutations operate on the AST, thus the edit distance between program token sequences of $P$ and $P'$ may be larger than the number of mutations, as shown in Figure 4. We defer more details on mutations to Appendix D. We generate an edit sequence to modify from $P'$ to $P$, then train the debugger with the standard cross-entropy loss using this edit sequence as supervision.

**Stage 2: Fine-tuning with the neural program synthesizer.** After pre-training, we fine-tune the model with the incorrect programs produced by the neural program synthesizer. Specifically, we run the decoding process using the synthesizer model on the Karel training set, then use those wrong predictions to train the debugger.

### 3.4 Inference Procedure

During inference, we achieve the best results using a best first search, as described in Algorithm 1. In the algorithm, we denote the synthesizer model as $M(e)$, which produces a list of candidate

---
**Algorithm 1** Best first search
---

1: **function** BEST-FIRST-SEARCH$_k$(($e$))
2:     $F \leftarrow M(e)$                        ▷ Frontier of the search space: programs yet to be expanded
3:     $S \leftarrow \{\}$                                                    ▷ Already expanded programs
4:     **for** $i \in \{1 \ldots k\}$ **do**
5:         $c \leftarrow \arg\max_{p \in F \backslash S} T(p, e)$
6:         $S \leftarrow S \cup \{c\}$
7:         **if** $T(c, e) = 1$ **then**
8:             **return** $c$                                                              ▷ Success
9:         **end if**
10:         $F \leftarrow F \cup D(c, e)$
11:     **end for**
12:     **return** $\arg\max_{p \in F} T(p, e)$    ▷ Probable failure, unless the program was found on the final step
13: **end function**

---

programs for input-output examples $e$. The debugger model $D(p, e)$ produces a list of candidate programs given the input program $p$. The function $T(p, e)$ executes the program $p$ on the examples $e$, and returns a value in $[0, 1]$ representing the proportion of input-output examples that are satisfied.

Within our SED framework, we view program synthesis from specification as a search on an infinite tree with every node except the root node being annotated with a program $c$, and having children $D(c, e)$. Our goal is to search for a program $p$ satisfying $T(p, e) = 1$. While $T(p, e) = 1$ does not ensure that the generated program is semantically correct, as $e$ does not include held-out test cases, it is a necessary condition, and we find it sufficient as the termination condition of our search process.

We design two search algorithms for SED. Our first algorithm is a *greedy* search, which iteratively selects the program from the beam output of the previous edit iteration that passes the greatest number of input-output examples (and has not yet been further edited by the debugger), and returns the edited program when it passes all input-output examples, or when it reaches the maximal number of edit operations allowed, denoted as $k$. See Algorithm 2 in Appendix C for more details.

A more effective scheme employs a *best-first* search. Compared to the greedy search, this search algorithm keeps track of all the programs encountered, as shown in line 10 of Algorithm 1, so that it can fall back to the debugger output from earlier edit iterations rather than get stuck, when none of the programs from the current edit iteration is promising.

## 4   Evaluation

In this section, we demonstrate the effectiveness of SED for Karel program synthesis and repair. We first discuss the evaluation setup, then present the results.

### 4.1   Evaluation Setup

The Karel benchmark [4, 2] is one of the largest publicly available input-output program synthesis dataset that includes 1,116,854 samples for training, 2,500 examples in the validation set, and 2,500 test examples. Each sample is provided with a ground truth program, 5 input-output pairs as the specification, and an additional one as the held-output test example. We follow prior work [2, 23, 3] to evaluate the following metrics: (1) **Generalization.** The predicted program $P$ is said to generalize if it passes the all the 6 input-output pairs during testing. This is the primary metric we consider. (2) **Exact match.** The predicted program is an exact match if it is the same as the ground truth.

**Program repair.** In addition to the Karel program synthesis task introduced above, we also evaluate our debugger component on the mutation benchmark in [25]. Specifically, to construct the test set, for each sample in the original Karel test set, we first obtain 5 programs $\{P_i'\}_{i=1}^5$ by randomly applying 1 to 5 mutations starting from the ground truth program $P$, then we generate 5 test samples for program repair, where the $i$-th sample includes $P_i'$ as the program to repair, and the same input-output pairs and ground truth program $P$ as the original Karel test set.

## 4.2 Synthesizer Details

We consider two choices of synthesizers. The first synthesizer is **LGRL** [2], which employs a standard encoder-decoder architecture as discussed in Section 3.1. During inference, we apply a beam search with beam size $B = 32$. We also evaluate a variant that performs the greedy decoding, i.e., $B = 1$, denoted as **LGRL-GD**. The second synthesizer is the execution-guided neural program synthesis model proposed in [3], denoted as **EGNPS**. The model architecture of EGNPS similar to LGRL, but it leverages the intermediate results obtained by executing partial programs to guide the subsequent synthesis process. During inference, we apply a search with beam size $B = 64$. We present the performance of these synthesizers in the first row ( "Synthesizer Only") of Table 1.

## 4.3 Debugger Details

We compare our debugger architecture incorporated with the trace embedding component to the baseline in [25], and we refer to ours and the baseline as *TraceEmbed* and *No TraceEmbed* respectively. For the program synthesis task, all models are pre-trained on the training set of the synthetic mutation benchmark with 1-3 mutations. For the fine-tuning results, LGRL and LGRL-GD are fine-tuned with their synthesized programs, as discussed in Section 3.3. For EGNPS, we evaluate the debugger fine-tuned with LGRL, because EGNPS decoding executes all partial programs generated in the beam at each step, which imposes a high computational cost when evaluating the model on the training set.

## 4.4 Results

**Mutation benchmark for program repair.** Figure 3 shows the results on the mutation benchmark. For each debugger architecture, we train one model with programs generated using 1-3 mutations, and another one with 1-5 mutations. Our most important observation is that the debugger demonstrates a good out-of-distribution generalization performance. Specifically, when evaluating on 4-5 mutations, although the performance of models trained only on 1-3 mutations are worse than models trained on 1-5 mutations, they are already able to repair around 70% programs with 5 mutations, which is desirable when adapting the model for program synthesis. On the other hand, each model achieves better test performance when trained on a similar distribution. For example, models trained on 1-3 mutations achieve better performance when evaluating on 1-3 mutations than those trained on 1-5 mutations.

Meanwhile, for each number of mutations, the lowest repair error is achieved by the model with our TraceEmbed component, demonstrating that leveraging execution traces is helpful. However, such models tend to overfit more to the training data distribution, potentially due to the larger model sizes.

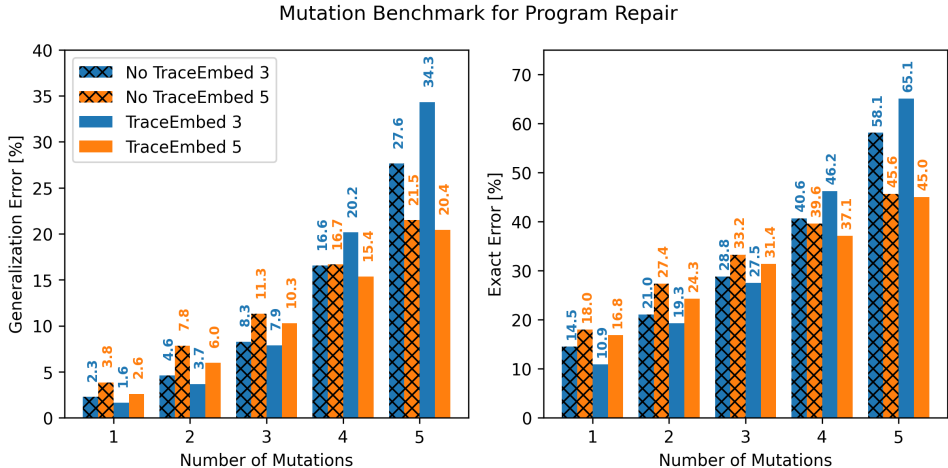

Figure 3: Results on the mutation benchmark, where x-axis indicates the number of mutations to generate the programs for repair in the test set. In the legend, "3" refer to models trained on 1-3 mutations, "5" refer to models trained on 1-5 mutations.

Table 1: Results on the test set for Karel program synthesis, where we present the generalization error with exact match error in parentheses for each synthesizer / debugger combination.

| Synthesizer+Debugger | LGRL-GD | LGRL | EGNPS |
|---|---|---|---|
| Synthesizer Only | 39.00% (65.72%) | 22.00% (63.40%) | 12.64% (56.76%) |
| No TraceEmbed+No Finetune | 18.56% (62.84%) | 14.88% (62.72%) | 10.68% (56.64%) |
| No TraceEmbed+Finetune | **16.12**% (**60.88**%) | 14.32% (**62.48**%) | **10.20**% (56.56%) |
| TraceEmbed+No Finetune | 18.68% (63.84%) | 14.60% (62.88%) | 10.68% (56.56%) |
| TraceEmbed+Finetune | **16.12**% (61.16%) | **14.28**% (62.68%) | 10.48% (**56.52**%) |

**The Karel program synthesis benchmark.** Table 1 presents our main results for program synthesis, where the debugger runs 100 edit steps. Firstly, SED consistently boosts the performance of the neural program synthesizer it employs. In particular, with LGRL-GD as the synthesizer, SED significantly outperforms LGRL without the debugger, which shows that the iterative debugging performed by SED is more effective than the standard beam search. Meanwhile, with EGNPS as the synthesizer, even if the synthesizer already leverages the execution information to guide the synthesis, SED still provides additional performance gain, which confirms the benefits of incorporating the debugging stage for program synthesis. Note that for EGNPS, we base our predictions on a single program synthesizer, rather than the ensemble of 15 models that achieves the best results in [3]. Even if we apply the ensemble as the program synthesizer for SED, which already provides a strong result of $8.32\%$ generalization error, SED is still able to improve upon this ensemble model to achieve $7.78\%$ generalization error.

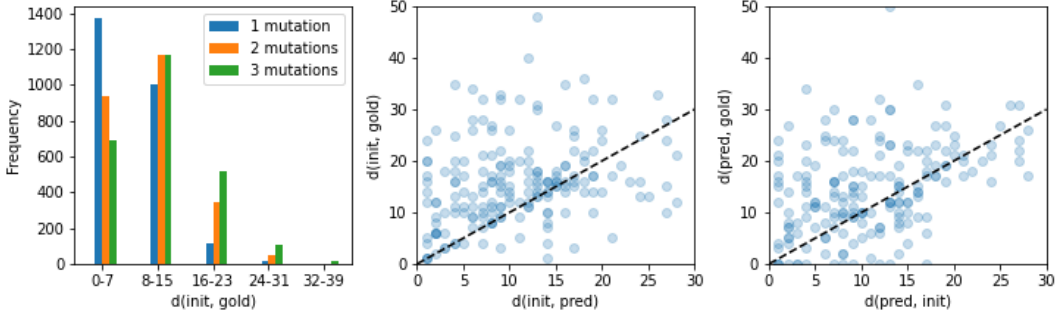

Figure 4: Left: The distribution of edit distances for the mutation benchmark by number of mutations. Middle and right: The joint distributions of the edit distances between the initial program predicted by the LGRL synthesizer (`init`), the `gold` program, and the program predicted by SED that passes all IO cases (`pred`). Dashed lines correspond to $x = y$.

To understand how SED repairs the synthesizer predictions, Figure 4 demonstrates the distribution of edit distances between the initial and ground truth programs in the pre-training dataset (leftmost), as well as distributions of edit distances among the ground truth, the predicted programs by the synthesizer, and the repaired programs by SED that are semantically correct (middle and right). The debugger is not fine-tuned, employs the trace embedding component, and performs 100 edit steps. Firstly, from the middle graph, we observe that SED tends to repair the synthesizer prediction towards a correct program that requires fewer edit steps than the ground truth, and we provide an example in Figure 1. The rightmost graph further shows that the repaired programs are generally closer to the initial predictions than to the ground truth, which could be the reason why SED achieves a much smaller improvement of the exact match than the generalization metric. Comparing these distributions to the leftmost graph, we note that without fine-tuning, SED is already able to repair not only program semantic errors that might not correspond to the synthetic mutations for training, but also programs with larger edit distances to the ground truth than the edit distances resulting from synthetic mutations, which again demonstrates the generalizability of SED.

Next, we discuss the effects of our trace embedding component and fine-tuning, and we further present the results with different number of edit steps in Figure 5. We observe that fine-tuning improves the results across the board, and has a particularly pronounced effect for LGRL-based models, where the data source for fine-tuning comes from the same synthesizer. Meanwhile, the

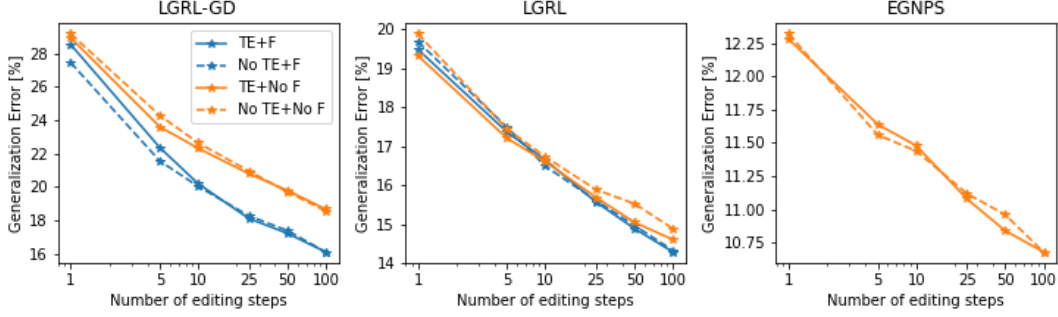

Figure 5: Comparison of different architectures and training process for program synthesis. TE refers to TraceEmbed, and F refers to fine-tuning on the data generated by the same synthesizer. Note the logarithmic scale of the $x$ axis.

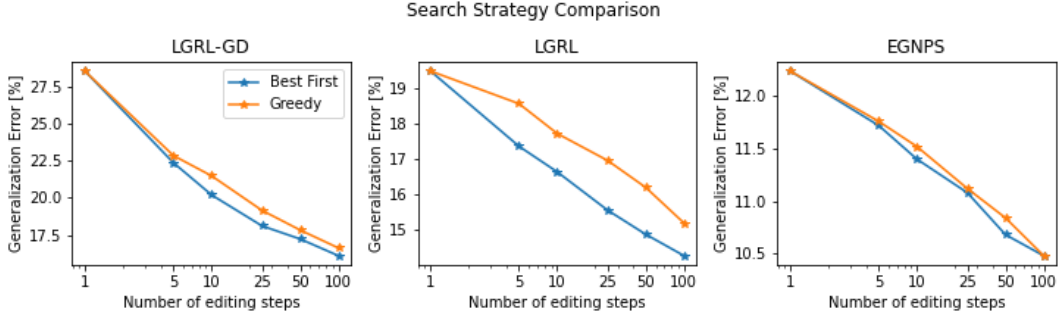

Figure 6: Comparison of best first and greedy search strategies. All models use TraceEmbed+Finetune as defined in Table 1.

Table 2: Generalization errors of the LGRL program synthesizer with SED and the standard beam search respectively, when expanding the same number of programs per sample.

| Debugger beam size | Edit steps | # of expanded programs | SED | Beam Search |
|---|---|---|---|---|
| 32 | 25 | 141 | **16.64%** | 18.52% |
| 64 | 25 | 229 | **15.72%** | 17.64% |
| 32 | 100 | 393 | **15.40%** | 17.00% |
| 64 | 100 | 687 | **14.60%** | 15.92% |

debugger accompanied with the trace embedding mostly achieves better performance, especially when fine-tuning is not performed. Note that the performance gain is not due to the larger model size. To confirm this, we train another model without `TraceEmbed`, but we increase the hidden sizes of `IOEmbed` and `ProgramEncoder` components, so that the number of model parameters matches the model with `TraceEmbed`. Using the LGRL synthesizer, this alternative debugger architecture achieves generalization errors of $15.88\%$ without fine-tuning and $16.64\%$ with fine-tuning respectively, which are even worse than their smaller counterparts without `TraceEmbed`. On the other hand, due to the small training set for fine-tuning, since the trace embedding component introduces additional model parameters, the debugger component could suffer more from over-fitting.

Figure 6 compares the best first and greedy search strategies. We see that best first search always outperforms greedy search, often being able to achieve a similar performance in half the number of edit steps. This effect is more pronounced for LGRL and EGNPS synthesizers, as they provide more than one program to start with, which best first search can more effectively exploit.

In Table 1, we demonstrate that with the same program synthesizer, SED provides a more significant performance gain than the standard beam search. In Table 2, we further compare the generalization errors of SED and the standard beam search, with the same number of expanded programs per sample. Specifically, using the LGRL program synthesizer, we evaluate SED with debugger beam sizes of 32 and 64 and edit steps of 25 and 100 respectively, and track the number of expanded programs per sample. Afterwards, we apply the standard beam search to the same LGRL program synthesizer, and we set the beam sizes to match the average number of expanded programs by SED.

Again, our results show that SED consistently outperforms the standard beam search, when the same number of programs are expanded.

## 5   Related Work

**Program synthesis from input-output examples.** Program synthesis from input-output specification is a long-standing challenge with many applications [16, 10, 19], and recent years have witnessed significant progress achieved by deep learning approaches [1, 20, 5, 2]. Different domain-specific languages (DSLs) have been investigated, such as AlgoLISP [12, 31] for array manipulation, FlashFill [20, 5, 29] for string transformation, and Karel [2, 23, 3] studied in this work. While most existing work only uses the execution results to post-select among a set of candidate programs predicted by a synthesizer, some recent work leverage more fine-grained semantic information such as the intermediate execution states to improve the synthesizer performance [27, 31, 3, 7]. In our evaluation, we demonstrate that SED further provides performance gain by leveraging execution results to repair the synthesized programs.

Besides neural network approaches, several generate-and-test techniques have been presented for program synthesis, mostly based on the symbolic search [22, 26]. Specifically, STOKE uses MCMC sampling to explore the space of program edits for superoptimization [22], while we demonstrate that with our neural network design, SED achieves good performance even with a simple heuristic search. CEGIS-based approaches utilize a symbolic solver for generating counterexamples to guide the program synthesis process [26, 13], which typically require a more complete and formal specification than a few input-output examples as provided in the Karel domain. We consider developing more advanced search techniques to improve the model performance as future work.

**Program repair.** There has been a line of work on program repair, including search-based techniques [15, 14, 18] and neural network approaches [11, 30, 25, 28, 6]. While most of these work focus on syntactic error correction, S3 designs search heuristics to improve the efficiency and generalizability of enumerative search over potential bug fixes, guided by input-output examples as the specification [14]. In [30], they train a neural network to predict the semantic error types for programming submissions, where they use execution traces to learn the program embedding. In [25], they study program repair on Karel where the wrong programs are generated with synthetic mutation, and we use its model as a baseline for our debugger component. Meanwhile, iterative repair is used as part of the decompilation pipeline in [9]. In this work, our SED framework incorporates the program repair scheme for input-output program synthesis, where the relationship between the program and the specification is typically complex.

Though both our work and [25] evaluate on Karel, there are several key differences. Most importantly, in [25], they didn't study how to incorporate a debugger component to improve the program synthesis results. Instead, they focus on repairing programs that are generated by randomly mutating the ground truth programs, thus they assume that the wrong programs are already syntactically similar to the ground truth. On the other hand, we demonstrate that the debugger component is not only helpful for the program repair task itself, but also improves the program synthesis performance. Furthermore, even if the program synthesizer generates programs syntactically far from the ground truth, the debugger may still be able to predict alternative programs that satisfy the specification.

## 6   Conclusion

Program synthesis and program repair have typically been considered as largely different domains. In this work, we present the SED framework, which incorporates a debugging process for program synthesis, guided with execution results. The iterative repair process of SED outperforms the beam search when the synthesizer employs the greedy decoding, and it significantly boosts the performance of the synthesizer alone, even if the synthesizer already employs a search process or incorporates the execution information. Additionally, we found that even though there is a program aliasing problem for supervised training, our two-stage training scheme alleviates this problem, and achieves strong generalization performance. Our SED framework could potentially be extended to a broad range of specification-guided program synthesis applications, and we consider it as future work.

## Broader Impacts

Program synthesis has many potential real-world applications. One significant challenge of program synthesis is that the generated program needs to be precisely correct. SED mitigates this challenge by not requiring the solution to be generated in one shot, and instead allowing partial solutions to be corrected via an iterative improvement process, achieving an overall improvement in performance as a result. We thus believe SED-like frameworks could be applicable for a broad range of program synthesis tasks.

## Acknowledgments and Disclosure of Funding

This material is in part based upon work supported by the National Science Foundation under Grant No. TWC-1409915, Berkeley DeepDrive, and DARPA D3M under Grant No. FA8750-17-2-0091. Any opinions, findings, and conclusions or recommendations expressed in this material are those of the author(s) and do not necessarily reflect the views of the National Science Foundation. Xinyun Chen is supported by the Facebook Fellowship.

## Footnotes

*Equal contribution. Work was done when Peter was visiting UC Berkeley.

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
