[Supplementary Material]

# A Hyperparameters

## A.1 LGRL Training

The LGRL model was trained with a learning rate of 1, that decayed by 50% every $10^5$ steps on 50 epochs of the Karel training dataset, using minibatch SGD with batch size 128 and gradient clipping with magnitude 1. It was run in greedy decoding mode to form the LGRL-GD synthesizer, and run using beam search with 32 beams to form the LGRL synthesizer.

## A.2 EGNPS Model

The EGNPS model was trained for 10 epochs on the Karel dataset, with a learning rate of $5 \times 10^{-4}$, and the batch size is 16. See [3] for more details. During the inference time, it was run in the search mode with a beam size of 64.

## A.3 Debugger Training

The debugger was trained with a learning rate of 1, that decayed by 50% every $10^5$ steps on 50 epochs of the Karel training dataset using random mutations, sampled with probability proportional to the number of mutations. Minibatch SGD was used with a batch size of 64, and gradient clipping with magnitude 1. The models were finetuned on examples from the training dataset that were incorrect, also for 50 epochs, with a learning rate of $10^{-4}$.

## A.4 TraceEmbed Architecture

The `TraceEmbed` unit is a residual convolutional network. The input, output, and trace grids are stacked along the channel axis, thus preserving locality in space while allowing features and which grid to be fully connected. The network is composed of an initial convolution that takes the $15 \times 3$ channels of the input grids to 64 channels, then three ResNet blocks, each of which consists two layers of of batch normalization, ReLU, and convolution followed by a residual connection. All convolutions are $3 \times 3$ with a padding of 1. The last layer is a fully connected layer that flattens the entire grid into an embedding of size $256$ (the same size as a program token embedding).

# B More Descriptions of the Karel Domain

Figure 7 presents the grammar specification of the Karel DSL. Specifically, the DSL describing its movements inside a grid consisting of cells which are of size 2x2 to 18x18 and containing between 0 to 10 objects. These movements are described with `move`, `turnLeft`, `turnRight` and interactions with the markers are `pickMarker` and `putMarker`. The language contains constructs with conditionals such while and for loops with `front`, `left`, `right`, `IsClear`, `markerspresent`, and negations. Each cell of the grid is represented as a 16-dimensional vector corresponding to the features described in Table 3.

```
   Prog p  ::=  def run() : s
   Stmt s  ::=  while(b) : s | repeat(r) : s | s₁ ; s₂ | a
            |   if(b) : s | ifelse(b) : s₁ else : s₂
   Cond b  ::=  frontIsClear() | leftIsClear() | rightIsClear
            |   markersPresent() | noMarkersPresent() | not b
 Action a  ::=  move() | turnRight() | turnLeft()
            |   pickMarker() | putMarker()
   Cste r  ::=  0 | 1 | ... | 19
```

Figure 7: Grammar for the Karel task.

# C Full Greedy Algorithm

The full greedy algorithm is in Algorithm 2.

| Robot facing North |
|---|
| Robot facing East |
| Robot facing South |
| Robot facing West |
| Obstacle |
| Grid boundary |
| 1 marker |
| 2 markers |
| 3 markers |
| 4 markers |
| 5 markers |
| 6 markers |
| 7 markers |
| 8 markers |
| 9 markers |
| 10 markers |

Table 3: Representation of each cell in the Karel state.

---

**Algorithm 2** Greedy search algorithm

---

1: **function** GREEDY-SEARCH$_k$($(e)$)
2:     $c \leftarrow \arg\max_{p \in M(e)} T(p, e)$
3:     $S \leftarrow \{\}$                                           ▷ Already expanded programs
4:     **if** $T(c, e) = 1$ **then**
5:         **return** $c$                                        ▷ Success
6:     **end if**
7:     **for** $i \in \{1 \ldots k\}$ **do**
8:         $c \leftarrow \arg\max_{p \in D(c,e) \setminus S} T(p, e)$
9:         $S \leftarrow S \cup \{c\}$
10:        **if** $T(c, e) = 1$ **then**
11:            **return** $c$                                 ▷ Success
12:        **end if**
13:     **end for**
14:     **return** $c$                                           ▷ Failure
15: **end function**

---

# D   Mutations

There are six types of mutations that we consider, identical to the ones used in [25]. Three mutations, $\texttt{insert}(n, a)$, $\texttt{delete}(n)$, $\texttt{replace}(n, a)$, each take a node $n$ and either delete it, replace it with some action $a$, or insert the action $a$ next to this node. The mutation $\texttt{wrap}(\bar{n}, t, c)$ wraps the series of nodes $\bar{n}$ in a control construct specified by the control type $t \in T_c$, where $T_c = \{\texttt{if, ifelse, while, repeat}\}$ and control value $c$, which is a conditional for if/while and a number for repeat. The $\texttt{unwrap}(n)$ mutation takes in a node whose root is a construct in $T_c$ and replaces it with its body. The mutation $\texttt{replaceControl}(n, c)$ takes a node $n$ whose root is a construct in $T_c$ and replaces the control value or number of repetitions with $c$, an appropriately typed control value. Each mutation maintains the syntactic validity of the program.