[Reviews · NeurIPS 2020]

Review 1

Summary and Contributions: This work proposes a framework called SED to augment neural program synthesizers with a “neural program debugger” component in order to repair the incorrect programs. If the synthesizer is unable to synthesize a correct program, the incorrect programs generated by the synthesizer are passed to the debugger. The neural programmer debugger iteratively modifies the programs to come up with a correct program. In addition to the syntactic information it also uses execution traces to effectively find the correct program. SED is evaluated on the Karel benchmark and outperforms a neural synthesizer even when restricted to a single beam.

Strengths: The idea of integrating program repair with program synthesis is simple but works pretty well. The results on the Karel dataset are impressive even when the synthesizer is restricted to just 1 beam. The proposed approach also improves the performance of the execution guided program synthesizer. Although the experiments are conducted in just one domain (Karel), they are thorough. There is in depth analysis of the results with respect to various variation of the synthesizer and the debugger configurations.

Weaknesses: * I would rate the current work medium on novelty. Program repair has already been explored separately in earlier work. Of course, making it work with program synthesizer is not trivial. * The approach has been shown to work in one domain (Karel). Replicating the same success on a different domain (possibly with more complex grammar) might be challenging. But this doesn’t take away the current contribution. * There are several trade-offs when designing a synthesizer. If execution traces are used, then there is a computational cost of evaluating the candidate programs. For this domain it may not be too expensive, but one can imagine grammar with functions that are too expensive to run often. Considering this, even in Karel, I would like to see the results (number of correct programs synthesized) within a given timeout. From the current results I cannot quantify the overhead imposed by the program repair mechanism. Update: Thank you for clarifying the difference with respect to [20]. I now understand that the focus of current work is on integration of the debugger with program synthesis and that the debugger can repair programs syntactically very different from ground truth programs. However, after discussion with other reviewers I feel that the approach is kind of underwhelming given that the components have been presented in the earlier work.

Correctness: The claims are sound and the experiments are well designed. As mentioned above, I would have liked some results on a different domain.

Clarity: The paper is well written and easier to follow along. I appreciate the detailed analysis of the results. A few minor comments. 1. Line 177: “Without Debugger”) of Table 1. Should be “Synthesizer only” 2. Figure 3 uses a circle fill pattern for NoTraceEmbed. This is harder to read. You may think of using a different pattern.

Relation to Prior Work: Related work has been discussed in sufficient details. However, I would like to see a detailed comparison with the work in [20] at a single place. I also would like the authors to compare their contributions with the following work which has many similarities with the current work. Le, Xuan-Bach D., et al. "S3: syntax-and semantic-guided repair synthesis via programming by examples." Proceedings of the 2017 11th Joint Meeting on Foundations of Software Engineering. 2017.

Reproducibility: Yes

Additional Feedback:


Review 2

Summary and Contributions: This paper extends prior work on specification-directed program repair (reference 20) with an explicit encoder for an execution trace, as well as a pre-training step (on synthetic mutations) before a training step on real, synthesized programs. Both additions seems to improve generalization accuracy mildly, for different synthesizers.

Strengths: TraceEmbed seems like a novel contribution (e.g., compared to execution-guided synthesis, which was using traces only for search). Generalization accuracy of debugging improves significantly over prior work. Pre-training on synthetic mutations before training on real repairs seems like a prudent strategy and has positive results.

Weaknesses: Although interesting as a contribution, TraceEmbed seems rather simple (a straightforward convolution) and has relatively limited positive effect. In fact, I suspect that the weights devoted to TraceEmbed would be put to better use increasing the hiddens of the IO and program encoders and decoder. Biggest reported benefit (the addition of the debugger) has been presented before in an ICLR workshop paper, but this is a more detailed presentation.

Correctness: I see no obvious flaws in the methodology or presentation. The experimental design seems to be done carefully. Some design-presentation flaws are described in the detailed comments.

Clarity: The paper is reasonably well written, except for casual grammatical errors. The presentation could improve from another round of simplification, e.g., with respect to notation and design illustrations (see detailed comments).

Relation to Prior Work: This paper extends an idea presented in an ICLR workshop a few years back. The differences are acknowledged in broad strokes.

Reproducibility: Yes

Additional Feedback: I enjoyed reading your paper. The notion of a debugger is intuitive and seems to have a very positive impact. Unfortunately, it's not a novel contribution, since it was published in the earlier ICLR workshop paper, which steals the thunder from this submission a bit. Pre-training on synthetic mutations, before fine-tuning on real buggy synthesized programs seems simple and effective! The contribution of TraceEmbed appears to be more of a negative result, given that it doesn't add that much, and it only improves accuracy before any real improvements have been made by the debugger. Overall, this is a nice extended presentation of the previously published concept of a neural debugger. Some details on the content: L26: I just reread the reference, and I saw no mention supporting this statement. The whole point of the reference is that partial-execution results improve on prior synthesizers that didn't use partial executions. So, I don't quite understand where you drew that conclusion from. I saw the author response to my puzzlement here, which said that a loop is treated the same no matter how many times it iterates, but still that doesn't support the statement that "partial programs are not always informative", but merely that it's not necessarily precise. Figure 2 is a little confusing. It looks like embedded inputs and outputs aren't going anywhere, but somehow appear as inputs to the edit decoder. Is that an attention operator? Regular inputs? Lines 105--110: It's confusing to have a $W$ set of $(u,t)$'s, and a $w$ weight that has all three indices $u, t, i$. Maybe use a different symbol for $W$? Line 109: If I'm understanding correctly, w_{u, t, i} is the same for any $u$ and $t$, since neither $u$ nor $t$ appears on its definition. Wouldn't it be sufficient to name it $w_i$ then? It might simplify your notation a bit. Line 111: What does the $ep_i || q_i$ notation mean? Is that concatenation? Logical disjunction? Figure 3 and Lines 196--197: "The lowest error is achieved by the model with our TraceEmbed component". This statement doesn't seem true. On 4 mutations, generalization error is 16.9% without TraceEmbed 3 but 20.3% with TraceEmbed 3. Similarly for 5 mutations. Did you forget to limit this statement to 1--3 mutations and TraceEmbed 5? Regardless of the bug in your text, why is this happening? Why is TraceEmbed harming the model when it's tested out of distribution? That seems puzzling. Figure 5: this result is rather disappointing. It says that TraceEmbed only benefits the debugger in the very beginning, and before multiple editing steps have had a chance to bring error down. The impressive drop in generalization error happens after multiple editing steps, and there TraceEmbed is essentially irrelevant. Therefore, I'd love to know if you were to use all those weights from the trace encoder to add more hiddens for the IO/Program encoder, would you get better results for the same model size? I read the response on this point and I appreciate the effort to explore the trade-off. It sounds like the result supports the EGNPS graph on Figure 5, which wasn't the issue, but for LGRL-GD the result is less clear. It would be helpful to know if the technique is meant to work with EGNPS, but not LGRL/LGRL-GD, or why, if that's the case, the benefit favors some synthesizers over others. Otherwise, the results are mixed.


Review 3

Summary and Contributions: SED addresses neural program synthesis from input-output examples by combining a neural synthesizer with a neural program editor in a generate-and-repair loop. After training the synthesizer model and the editor model (on synthetic program edits + fine-tuned on real model errors), the framework searches for a sequence of edits applied to the predicted program that would turn it into a correct one (satisfying all examples). The search is either greedy or best-first with a frontier. The approach is evaluated on Karel with off-the-shelf SOTA synthesizer and editor models, and it significantly improves upon baseline prediction accuracy, although it's unclear whether it reaches SOTA.

Strengths: + A combination of neural synthesis and neural editing in a generate-and-repair loop is natural. Its analogues exist in symbolic program synthesis and in other DL applications. + The approach significantly improves upon the baselines (synthesizer only, with greedy or beam inference). + The work is appropriate and relevant to the NeurIPS program synthesis community.

Weaknesses: - The approach is underwhelming: it takes existing components, combines them in a loop, and applies a textbook best-first search. - The baselines are not trained strongly enough, the same model architectures report higher SOTA numbers in their proposed works. In fact, SED only closes the gap between previously reported numbers and the presented baseline ones.

Correctness: The methodology is sound. Training a neural editor on the error done by a specific neural synthesizer model and then testing it on the same model could be perceived as cheating. It is actually perfectly valid – the editor is not intended to generalize to other models, and the two in tandem are effectively just a disentangled version of a two-pass autoregressive model. But this could confuse a reader, so worth calling out explicitly. (In fact, I found EGNPS experiments the most interesting, because in there the editor is *not* trained on the same model yet still corrects a significant number of mistakes from a different distribution.)

Clarity: The paper is generally straightforward and well written. Minor exceptions: - Eqs 1-2 and text around them are unnecessarily complicated. The weight on each edge is the same within its corresponding target group. Why not just write "mean" in Eq 2 and avoid the w_{u,t,i} notation altogether? - In Eq.2, e_{u,t} should probably be te_{u,t}. - Concatenation is sometimes denoted as [·;·] and sometimes as [·||·]. - L207-219 need proofreading. The sentences are long, clunky, require several passes to parse, and ungrammatical in a few places. - Algorithm 1: "Fringe" is usually called "frontier" in the search literature.

Relation to Prior Work: Relation to prior work on neural program synthesis and editing is clear and fairly discussed. Generate-and-test is also common in symbolic program synthesis, starting with CEGIS [1]. It does not necessarily edit the candidate program but rather generates a counterexample for it. However, editing-based synthesis approaches exist too, e.g. [2]. Please compare to the relevant literature (not experimentally). In the neural program synthesis world, CEGIS can be similarly "automated" by a loop of two models + search [3]. [1] Solar-Lezama, A. and Bodik, R., 2008. Program synthesis by sketching (p. aAI3353225). University of California, Berkeley. [2] Schkufza, E., Sharma, R. and Aiken, A., 2013. Stochastic superoptimization. ACM SIGARCH Computer Architecture News, 41(1), pp.305-316. [3] Laich, L., Bielik, P. and Vechev, M., 2019, September. Guiding Program Synthesis by Learning to Generate Examples. In International Conference on Learning Representations.

Reproducibility: Yes

Additional Feedback: I'm borderline on the paper. On one hand, this is a natural combination of two recent lines of research – execution-guided neural program synthesis and neural program repair. It clearly shows significant improvement over neural synthesis alone. On the other hand, the combination is underwhelming: a) the synthesis and editing models are mostly off-the-shelf, with a notable exception of adding a form of execution-guidance to the editor. b) the search is just textbook best-first search, missing numerous guided-search opportunities from the past 10 years. Integration of EG into a neural editor model is novel and interesting. I have a few questions on it, however: - The "alignment" between the program and its trace is a bit imprecise and heuristic. It seems to model program dependency tracing literature ("which program instructions contributed to this trace action"). However, it's inconsistent: a loop header is aligned to every action resulted from its corresponding loop body, but an `if` conditional guard is not aligned to its two branches. - The embeddings of all the actions for the same target are averaged, which smoothes over a lot of relevant information. Have you tried other forms of pooling? - If you define an alignment graph and pool information from one into the other along the edges, why not go all the way and model the whole thing as a GNN? The proposed best-first search improves upon greedy only by ≤1% by the end, even if one reaches the same accuracy twice as fast. Moreover, the improvement curve for both has roughly the same slope on the logarithmic axis, so the improvement quickly plateaus as the number of edits increases. This raises the question — why best-first search? The closest similar work, STOKE [1], explores the complex space of program edits using MCMC with Metropolis-Hastings. Its proposal distribution is manual rather than learned, but learned approaches exist too [2]. One could also explore the same space using MCTS. Or learned MCTS (AlphaGo). More generally, fine-tuning the editor on a supervised dataset of synthesis errors but then using it blindly in a simple search seems like underutilizing resources and ignoring a plethora of research on searching complex spaces shaped by the model decisions. In the evaluation, the reported baseline accuracy for EGNPS (80.6%) is *much* lower than the reported single-model accuracy in the EGNPS paper (91.68%). The proposed SED method brings the accuracy to 88.64%, still lower than SOTA. Could you please explain why that's the case? To be clear, a method doesn't need to reach SOTA to be accepted if it's a valuable contribution, but it raises the question whether SED would be as effective against a strong baseline (whose error distribution might not be straightforward to model). Questions: - What's the beam size for EGNPS? - If you run beam search on the synthesizer with the beam size equivalent to the number of search iterations, how often would you find the program finally obtained by SED search in it? I wonder if the synthesizer and editor are learning the same program distribution. [1] Schkufza, E., Sharma, R. and Aiken, A., 2013. Stochastic superoptimization. ACM SIGARCH Computer Architecture News, 41(1), pp.305-316. [2] Wang, T., Wu, Y., Moore, D. and Russell, S.J., 2018. Meta-learning MCMC proposals. In Advances in neural information processing systems (pp. 4146-4156).


Review 4

Summary and Contributions: This paper presents a program synthesis (i.e., software program generation) approach called “synthesize, execute, debug” (SED). The novelty of the approach, according to the authors is in the debugging aspect of program synthesis, where it attempts to repair a synthesized program that does not properly solve the provided input/output pairs.

Strengths: The idea of “repairing” an incorrect program could be useful in many real-world scenarios. For example, such a technique could be used to automatically fix bugs. As such, the idea of a synthesis system that can “debug” and “repair” a broken program could have notable utility, in my opinion.

Weaknesses: Unfortunately, from my perspective, there are many weaknesses in this paper in its current form, two of which are fairly major. When considered holistically, it is my opinion that this paper is not acceptable for NeurIPS publication at this time. The first reason for rejection is that this paper seems largely similar to the prior work of Shin et al. “Towards Specification-Directed Program Repair”, which was presented at an ICLR workshop in 2018. Many aspects of Shin’s ICLR 2018 work are seemingly identical to this work. Two such examples are: (1) the choice of the programming language (i.e., Karel) and (2) the idea of “program repair” -- renamed as “debug” in this paper -- but functions with nearly identical semantics to Shin’s 2018 system. However, there are improvements and differences from Shin’s 2018 system. Yet, they seem largely incremental, in my opinion. In short, it’s hard for me to argue that this paper offers notable insight over Shin’s ICLR workshop paper in 2018. The exception is a slight improvement of performance via various neural backends. The second reason for rejection, which is the core reason why I believe this paper is not acceptable for publication at __any__ tier-1 research venue is its utilization of the programming language Karel as a vehicle to demonstrate program repair. In short, Karel is, as noted by its creator Richard Pattis, a programming language meant for beginners and educational purposes only. It is not meant for any real world programming. Thus, the demonstration of program repair using Karel programs seems largely like an academic exercise to me. Given the simplicity of the language (e.g., it only contains five unique instructions), it’s unclear to me how program repair illustrated in Karel can provide forward progress for research in this space or more generally how the proposed approach might be applicable for real-world scenarios or programming languages, all of which have orders of magnitude more instructions than Karel. Due to this lack of real-world value and my inability to see how such research might generalize or be applicable for real-world problems, I’m unable to find an angle where I can argue for this paper’s acceptance. It would likely be a fine workshop paper (perhaps even a NeurIPS workshop paper?) but, I’m sorry to say that it seems unacceptable to me as a conference research publication and falls far short of my bar for NeurIPS publications. If the authors could demonstrate their “program repair” technique in real-world programs using real-world programming languages (e.g., JavaScript, Python, C/C++, Go, Java, etc.), I believe the paper and research would be likely be notably more interesting and significantly more convincing. Moreover, such an application would likely provide insight into whether the approach has true potential for real-world application. As the paper stands now, I do not believe it provides such insight. UPDATE: In my discussion with other reviewers and the meta-reviewer I have been convinced that my perspective on Karel has been too narrow. As such, I am increasing my reviewer score. While it seems my original rating was unfairly negative, I still can't justify giving this paper an accept due to the following (real or perceived) limitations: - limited novelty over prior workshop paper (perhaps this is just a perception, but perhaps this could be more clearly explained so future reviewers don't struggle with this as I did?) - limited experimental evaluation (although admittedly, perhaps that's a bias of mine related to Karel, which I'm trying hard to eliminate) or evaluation against more synthesizers - limited intuition about why certain decisions were made (I found the ICLR workshop paper to be much easier to understand, overall) Overall, to reiterate, I rather like the idea of automated program repair / debugging / fixing. The general concept, I believe has tremendous value. So, I hope that my negative review isn't perceived as this work not being worth pursuing. I absolutely think it is important research. Unfortunately, I think the paper in its current form isn't ready for a tier-1 venue such as NeurIPS.

Correctness: They appear to be.

Clarity: The writing is clear.

Relation to Prior Work: Not exactly. It is largely related to Shin et al.'s "Toward Specification-Directed Program Repair" an ICLR 2019 workshop paper. And, unfortunately, many of the decisions made by Shin et al. in their 2018 paper were not discussed in this paper, making some decisions seem arbitrary or random.

Reproducibility: Yes

Additional Feedback:

[Author Response · NeurIPS 2020]

Table 1: Generalization errors of SED and LGRL with the same number of expanded programs per sample.

| Debugger beam size | Debugger edit steps | # of expanded programs | SED | LGRL |
|---|---|---|---|---|
| 32 | 25 | 141 | **16.64%** | 18.52% |
| 64 | 25 | 229 | **15.72%** | 17.64% |
| 32 | 100 | 393 | **15.40%** | 17.00% |
| 64 | 100 | 687 | **14.60%** | 15.92% |

We thank reviewers for constructive comments! We first address the common confusion, then the individual questions.

**Differences with [20].** We would like to clarify that in [20], they didn't study how to incorporate a debugger component
to improve the program synthesis results. Instead, they focus on repairing programs that are generated by randomly
mutating the ground truth programs, thus they assume that the wrong programs are already syntactically similar to the
ground truth. On the other hand, we demonstrate that the debugger component is not only helpful for the program
repair task itself, but also improves the program synthesis performance. In particular, even if the program synthesizer
generates programs that are syntactically far from the ground truth, the debugger may still be able to predict alternative
programs that satisfy the specification. We will add a more detailed discussion in our revision.

**Comparison of SED and the synthesizer with the same execution budget.** Based on R1 and R3's questions, we
compute the number of programs expanded by SED, and run the synthesizer with the same beam size. As shown in
Table 1, SED still achieves better results than the synthesizer in this case. We will discuss more details in our revision.

**R1.** See the common response about the comparison with the synthesizer with the same execution budget, and the
comparison with [20]. We will discuss the related work in our revision. Specifically, S3 focuses on designing search
heuristics to improve the efficiency and generalizability of enumerative search over potential bug fixes, without any
learning techniques. SED instead incorporates a neural debugger into the neural program synthesis framework.

**R2.** To show that the effect of TraceEmbed is not just due to the increase of the model size, as suggested by R2, we
added experiments to increase the hidden size of IOEmbed and program encoder without TraceEmbed, so that the
numbers of parameters for models with/without TraceEmbed are similar. However, the results are even worse than the
smaller counterpart; e.g., with EGNPS, the generalization error is 13%, compared to 11.52% in the submission.

See the common response for the comparison with [20]. For L26, according to Table 2 in [3], the statements in a While
loop are only runned once during the While body prediction, and all remaining iterations are executed after finishing
the While loop generation, thus the partial execution results of a While body are essentially the same as an If-statement.
In Figure 2, the embedded IOs are inputs to both TraceEmbed and EditDecoder, and we use the same colors to denote
the same embeddings. $ep_i||q_i$ means the concatenation. About Lines 196–197, we mean that comparing the results of
all 4 models, the lowest error is achieved by the model with TraceEmbed and trained on similar mutations. In Lines
197-198, we discussed that a potential reason why TraceEmbed harming the model when it's tested out of distribution
may be the larger model sizes, which could lead to overfitting. We will clarify these points in our revision.

**R3.** Note that EGNPS achieves 91.68% generalization accuracy with an ensemble of 15 models, instead of a single
model. We added experiments to build upon EGNPS with a higher accuracy, and SED also further improves the results.
For example, using a single EGNPS synthesizer, the generalization accuracy is 89.52%, higher than 86.04% in the
EGNPS paper. With the ensemble as the synthesizer, SED still improves the accuracy by 0.52%.

An "if" conditional guard is also aligned to its two branches. About different forms of pooling and using a GNN, we
tried learning an attention map to aggregate the embeddings of actions, adding more types of graph edges, etc. However,
these variants are significantly slower to train and do not perform well, so we did not include them in our submission.

Studying more advanced search techniques is not the main focus of this work, but we agree that this is a promising future
direction. The fact that SED works even with a simple search justifies the overall design of our framework. We will
provide more discussion about work on generate-and-test program synthesis. Specifically, Schkufza et al. performed
MCMC sampling for superoptimization, while we train a neural network for program synthesis from input-output
specifications. Laich et. al. and Solar Lezama et. al. propose counter-example generation approaches that require
symbolic solvers, while we focus on program synthesis with a limited number of input-output examples as specification.

**R4.** See the common response about the differences with [20]. We would like to clarify that our main focus is not
the program repair task itself. Instead, we propose to integrate a neural program repair component to improve the
performance of a neural program synthesizer. The reviewer questions whether evaluation on Karel is sufficient to
demonstrate the effectiveness of our approach. We would like to note that most existing work on input-output program
synthesis was evaluated on domain-specific languages (DSLs), such as Karel (e.g., [2, 3, 4, 18]) and FlashFill (e.g., [5,
10, 16, 23]), and Karel is already among the most full-fledged DSLs with conditional and loop semantics. Therefore,
while we agree that evaluating on more popular programming languages could strengthen this work, we believe our
work is pushing forward along one direction that the neural program synthesis community is working on.

[Meta-Review · NeurIPS 2020]

The reviewers gave mixed scores and raised various issues with respect to the evaluation and the novelty compared to the ICLR 2018 workshop paper. However, I felt that accepting the paper at this time would be valuable to the NeurIPS community as it combines pre-existing components in a reasonable way, yielding both good performance improvement and providing useful insights about these models. I would encourage the authors to address the issues raised by the reviewers in the camera-ready.